# Periparturient Mineral Metabolism: Implications to Health and Productivity

**DOI:** 10.3390/ani14081232

**Published:** 2024-04-19

**Authors:** Achilles Vieira-Neto, Ian J. Lean, José Eduardo P. Santos

**Affiliations:** 1Department of Animal Sciences, University of Florida, Gainesville, FL 32611, USA; achillesneto.vet@gmail.com; 2Scibus, Camden, NSW 2570, Australia; ianl@scibus.com.au; 3Faculty of Veterinary Science, The University of Sydney, Camden, NSW 2570, Australia; 4DH Barron Reproductive and Perinatal Biology Research Program, University of Florida, Gainesville, FL 32611, USA

**Keywords:** calcium, dairy cow, dietary cation-anion difference, mineral

## Abstract

**Simple Summary:**

Mineral metabolism is altered with the onset of lactation, and a reduction in concentrations of calcium (Ca) in blood, particularly when it persists for days, is associated with increased risk of peripartum diseases. Hypocalcemia affects immune function, which seems to underlie the susceptibility to other diseases. Hypocalcemia is not caused by inadequate Ca intake but by the inability to adapt to the irreversible losses of Ca that disrupt homeostatic mechanisms that control blood Ca concentrations. A common method to reduce hypocalcemia is the feeding of acidogenic diets prepartum. Alternative strategies such as prepartum feeding P and Ca chelating agents and administration at calving of active vitamin D_3_ metabolites have shown promising results.

**Abstract:**

Mineral metabolism, in particular Ca, and to a lesser extent phosphorus (P) and magnesium (Mg), is altered with the onset of lactation because of extensive irreversible loss to synthesize colostrum and milk. The transient reduction in the concentration of Ca in blood, particularly when it lasts days, increases the risk of mineral-related disorders such as hypocalcemia and, to a lesser extent, hypophosphatemia. Although the incidence of clinical hypocalcemia can be reduced by prepartum dietary interventions, subclinical hypocalcemia remains prevalent, affecting up to 60% of the dairy cows in the first 3 d postpartum. More importantly, strong associations exist between hypocalcemia and increased susceptibility to other peripartum diseases and impaired reproductive performance. Mechanistic experiments have demonstrated the role of Ca on innate immune response in dairy cows, which presumably predisposes them to other diseases. Hypocalcemia is not related to inadequate Ca intake as prepartum diets marginal to deficient in Ca reduce the risk of the disease. Therefore, the understanding of how Ca homeostasis is regulated, in particular how calciotropic hormones such as parathyroid hormone and 1,25-dihydroxyvitamin D_3_, affect blood Ca concentrations, gastrointestinal Ca absorption, bone remodeling, and renal excretion of Ca become critical to develop novel strategies to prevent mineral imbalances either by nutritional or pharmacological interventions. A common method to reduce the risk of hypocalcemia is the manipulation of the prepartum dietary cation-anion difference. Feeding acidogenic diets not only improves Ca homeostasis and reduces hypocalcemia, but also reduces the risk of uterine diseases and improves productive performance. Feeding diets that induce a negative Ca balance in the last weeks of gestation also reduce the risk of clinical hypocalcemia, and recent work shows that the incorporation of mineral sequestering agents, presumably by reducing the absorption of P and Ca prepartum, increases blood Ca at calving, although benefits to production and health remain to be shown. Alternative strategies to minimize subclinical hypocalcemia with the use of vitamin D metabolites either fed prepartum or as a pharmacological agent administered immediately after calving have shown promising results in reducing hypocalcemia and altering immune cell function, which might prove efficacious to prevent diseases in early lactation. This review summarizes the current understanding of Ca homeostasis around parturition, the limited knowledge of the exact mechanisms for gastrointestinal Ca absorption in bovine, the implications of hypocalcemia on the health of dairy cows, and discusses the methods to minimize the risk of hypocalcemia and their impacts on productive performance and health in dairy cows.

## 1. Introduction

One of the consequences of lactation is the change in mineral metabolism, in particular, Ca. Colostrum is rich in Ca, with approximately 2.0 to 3.0 g/L [1,2,3]. In Holstein cows, 20 to 40 g of Ca are secreted by the mammary gland on the first day postpartum [1,2,3], and some cows are unable to properly adapt to this abrupt and irreversible loss of the mineral, thus impairing Ca homeostasis and leading to the development of hypocalcemia. Immediately before calving, a Holstein cow requires approximately 20 to 30 g/d of absorbable Ca, 9 to 10 g for uterine tissue accretion [4], 10 g to replenish the daily fecal losses when consuming 10 to 11 kg of dry matter daily [5], and another 1.5 to 10 g to meet the urinary losses depending on the type of diet fed [3,6]. With colostrogenesis, the daily needs for absorbable Ca increase substantially, and Ca sequestration by the mammary gland starts before parturition [7], thus reducing concentrations in plasma at least 9 h before calving [8]. In some cases, such as multiparous Jerseys fed an alkalogenic diet, the decline in plasma Ca concentration seemed to start the day before calving [9].

Normocalcemic dairy cows maintain plasma total Ca in a range of 2.2 to 2.5 mM. Of the total Ca in plasma, fractions vary with pH and albumin content, but approximately 48 to 52% is biologically active in the ionized form (Ca^2+^), 40 to 45% is bound to proteins, mainly albumin, with the remainder bound to soluble anions such as bicarbonate, phosphate, citrate, or lactate. Because of the sudden change in Ca needs, it is not surprising that the prevalence of dairy cows with serum Ca < 2.0 mM in the first 48 h after calving is at least 25% in primiparous and 45% in multiparous cows [10]. Nevertheless, the true incidence of subclinical hypocalcemia in early lactation depends on the frequency of blood sampling and the threshold used for the concentration of total Ca in serum or plasma [11]. Although different thresholds have been used, the most commonly cited is total Ca less than 2.0 mM [10], which is easily justified as almost every healthy cow past early lactation maintains plasma total Ca above 2.0 mM. When sampling of blood is daily, then incidence ranges from 40 to 60%, even when cows are fed diets designed to improve peripartum Ca metabolism [2,12]. When concentrations of Ca in the blood drop, both total Ca and Ca^2+^ decrease, which has consequences to dry matter intake (DMI), energy metabolism, and immune function [13,14], thereby predisposing cows to other periparturient diseases [15,16,17]. This review provides insights into the current understanding of mineral metabolism in periparturient dairy cows with a focus on Ca and its impacts on health and productivity, and discusses nutritional and pharmacological strategies to reduce the risk of hypocalcemia.

## 2. Risk Factors for Periparturient Mineral Imbalance

Colostrogenesis and the onset of lactation are likely the most important risk factors for hypocalcemia, as removal of the mammary gland prevented cows from experiencing a decrease in blood Ca at parturition [9]. Interestingly, mastectomy did not prevent the reduction of blood P, possibly because the maintenance of blood P depends on the consumption of dietary P and rumination to recycle P to the gut through saliva, both of which are affected by parturition, which suggests that the etiology of hypocalcemia differs from that of hypophosphatemia during the periparturient period.

DeGaris and Lean [17] found that the risk of milk fever increased by 9% per lactation. The increased risk may be ascribed to greater colostrum yield, as there was a 50% increase in the amount of Ca secreted in colostrum, from 16.9 in primiparous to 25.5 g in multiparous cow at the first postpartum milking [3]. This marked increase in irreversible loss of Ca requires immediate homeostatic mechanisms to ensure that blood total and Ca^2+^ will not drop below critical values that impair cellular function [13,14]. The need for an immediate homeostatic response is emphasized by a limited plasma pool of 2 to 4 g of Ca and the lag time between stimulation of parathyroid hormone (PTH) and calcitriol-induced uptake of Ca from the apical epithelium of the gastrointestinal tract (Figure 1). Further, readily labile bone reserves of Ca are limited, perhaps as little as 8 to 10 g [18].

In addition to the increased loss of Ca in colostrum, intestinal vitamin D receptor (VDR) abundance differs substantially between calves at 1 to 2 months of age and lactating cows of at least 9 years old [19]. Such changes in VDR at the apical membrane of the rumen epithelium may explain some of the increased risk of milk fever as cows age; however, a systematic description of the distribution and abundance of VDR in the gastrointestinal tract of Brown Swiss and Holstein cows showed no difference between those with a mean age of 3.6 years, lactations 1 and 2, and cows with a mean age of 6.9 years, lactations 3 to 6 [20]. Old cows, in particular, Jerseys, which have increased susceptibility to hypocalcemia, may also have reduced vitamin D-stimulated absorption of Ca in the gastrointestinal tract. Nevertheless, dairy cows in most farms typically are younger than 7 years of age, and the current data suggest that VDR expression might not be a reason that underlie increased susceptibility to hypocalcemia as cows move from lactations 1 to 4 [10].

In contrast, as rats age, both bone and intestinal VDR abundance decrease [19]. Older rats have less capacity to synthesize 1,25-dihydroxyvitamin D_3_ because of reduced PTH receptors in kidney cells [21]. Moreover, older rats have reduced expression of Ca-binding proteins that play a role in the transcellular transport of Ca in the intestine and kidney [22].

Although the concentration of Ca in serum in the first 2 d postpartum decreased as parity increased, the largest increment in risk of hypocalcemia as cows aged was observed between the first and second lactation [10]. It is unknown if the ability of vitamin D to facilitate gastrointestinal absorption of Ca or influence bone remodeling changes as cows age from first to second lactation. There is evidence that primiparous cows have more active bone remodeling during the transition period than multiparous cows [3]. The latter data suggest that the pattern of changes in Ca homeostasis as animals age is likely to occur in dairy cows, despite modern dairy cows having a mean lifespan of only 6 to 7 years in most farms. In any case, the loss of Ca with colostrogenesis and onset of lactation is likely the major cause of hypocalcemia [9], and multiparous cows have increased risk because they secrete 50% more Ca than primiparous cows [3]. Therefore, it is suggested that one of the main reasons for the increased risk of hypocalcemia as cows age is the increased loss of Ca in colostrum concurrent with potential alterations in bone remodeling, although one cannot discard reduced gastrointestinal Ca absorption.

Dairy cows fed diets with high potassium (K) and inadequate Mg contents have impaired ruminal absorption of Mg, predisposing them to hypomagnesemia [23,24]. Evidence suggests that tissue responsiveness to PTH is dependent on Mg [25], as responses to PTH in target tissues rely on the activation of adenylate cyclase [26] and phospholipase C [27], enzymes that require Mg for adequate cellular response [28,29]. In addition, hypomagnesemia reduces the ability of the parathyroid gland to secrete PTH in humans, even in the presence of hypocalcemia [30], and it is possible that similar mechanisms might also be present in dairy cows. Although experiments have not been conducted to titrate the adequate Mg content in prepartum diets, calculations have been used to suggest a dietary concentration of Mg that results in ruminal fluid concentrations that induce paracellular flow down the concentration gradient. Goff [15,31] indicated that when dietary Mg content is in low concentration, less than 0.25% of the diet dry matter, thereby resulting in a daily intake of 25 to 30 g in a prepartum cow consuming 10 to 12 kg of dry matter, the ruminal fluid concentrations of ionized Mg would be insufficient to induce paracellular transport, and absorption would rely primarily on the active transcellular transport mechanisms. In ruminants, the rumen-reticulum is the site for Mg absorption [24] and bypassing the forestomach-induced hypomagnesemia in sheep [32]. The apparent digestibility of Mg in dairy cows decreases with increasing dietary K content [33]. At 1% dietary K, the apparent digestibility of dietary Mg was estimated at 24.1%, whereas at 2% dietary K, the digestibility of Mg decreased to 16.6% [33]. The epithelium of the rumen papillae has an electrical potential difference of −40 to −70 mV. A high intake of K increases ruminal concentrations, which depolarizes the apical membrane of rumen papillae, thus perturbing the transepithelial Mg transport across the cell [34]. On the other hand, at high dietary Mg, concentrations of ionized Mg increase sufficiently to induce paracellular transport. In fact, absorption of dietary Mg is influenced by the intake of Mg in diets with high, but not low K [35]. In wethers, when dietary K was 1% of the diet dry matter, increasing dietary Mg from 0.13 to 0.37% did not affect the percentage of Mg that was absorbed, which was approximately 35%, although grams of Mg absorbed increased; on the other hand, at high dietary K content, 3.6% of the diet dry matter, increasing dietary Mg from 0.13 to 0.37% of the diet dry matter increased the percentage absorbed from 16.5 to 24.0% [35]. At the typical rumen concentration of ionized Mg of 0.2 to 0.4 mM, most absorption is through the transepithelial transport; however, at concentrations above 0.4 mM, paracellular diffusion is expected to be responsible for a considerable portion of Mg absorption [15,34]. Low dietary Mg might predispose cows to hypocalcemia [36], and it has been suggested that blood Mg concentration below 0.65 mM increases the risk of hypocalcemia [37].

A high intake of dietary P might also predispose cows to Ca imbalance in the peripartum period. A mathematical model predicted that increasing dietary P content from 0.3 to 0.4% of diet dry matter prepartum would result in an 18% greater risk of clinical hypocalcemia [36]. Dietary P content has been shown to linearly increase blood P in dairy cows [38], and in mouse models, increased blood P stimulates osteocytes to synthesize the bone-derived hormone fibroblast growth factor 23 (FGF23) [39], a mechanism that presumably is present in bovine although confirmation is needed. The main role of FGF23 is to control blood P and prevent hyperphosphatemia through two mechanisms. First, FGF23 reduces the reabsorption of P by reducing the expression of the sodium-phosphate cotransporters NaPi-2a (SLC34A1) and NaPi-2c (SLC34A3) in the cortex of the kidney [40], leading to increased urinary output of P. Secondly, FGF23 regulates the rate-limiting enzyme in vitamin D metabolism, 1α-hydroxylase (CYP27B1), responsible for the conversion of 25-dihydroxyvitamin D_3_ to 1,25-dihydroxyvitamin D_3_ in the kidney [41]. Dairy cows fed diets with very low P content, 0.15% prepartum and 0.20% postpartum, had increased plasma Ca concentrations compared with cows fed diets with 0.28 and 0.44% P pre- and postpartum, respectively, suggesting a possible role of dietary P on peripartum Ca homeostasis [42]. Cows fed the diets limited in P had less inorganic P in plasma, which possibly prevented the downregulation of 1α-hydroxylase by FGF23. Although differences were numerical, cows fed diets limited in P had decreased concentrations of PTH and increased concentrations of 1,25-dihydroxyvitamin D_3_ [42], perhaps suggesting an increased sensitivity to PTH to maintain Ca homeostasis around calving. Some evidence from a controlled experiment exists that limiting dietary P prepartum benefits peripartum Ca metabolism [43], and systematic reviews of the literature suggest an association between increasing dietary P prepartum and increased risk of hypocalcemia in early lactation [36,44]. In addition to the calcemic effects, FGF23 also inhibits 1α-hydroxylase in peripheral blood mononuclear cells [45], which might influence the suggested immunomodulatory effects of vitamin D [46]. Appropriate dietary contents of K (<1.20% of the diet dry matter), P (<0.30% of the diet dry matter), and Mg (0.35 to 0.45% of the diet dry matter) in prepartum diets are critical to improve Ca homeostasis during the periparturient period and reduce the risk of hypocalcemia [15].

Differences in susceptibility to hypocalcemia exist across different breeds, although limited mechanistic data are available to explain those findings. Jersey cows have 2.4 times greater odds of developing milk fever compared with Holstein cows [36]. Roche and Berry [47] reported that, in grazing production systems, Jerseys had five times the odds of developing milk fever compared with Holsteins, whereas crossbreds Holstein and Jersey were at intermediate odds of having milk fever. It is unlikely that management would explain the differences observed between those breeds. On the other hand, most of the nutritional recommendations for periparturient dairy cows are based on experiments using Holsteins. It is possible that maintenance of Ca homeostasis around calving differs between Jerseys and Holsteins, and the nutritional approaches to prevent hypocalcemia might not be exactly the same between the two breeds.

Body condition has been associated with hypocalcemia, and over-conditioned cows are at greater risk of having the disease [48]. Indeed, over-conditioned cows had increased milk yield in response to oral Ca dosing after calving [49] and improved health performance in response to exogenous administration of 1,25-dihydroxyvitamin D_3_ at parturition, which drastically reduced the prevalence of hypocalcemia [50]. Because excessive BCS is associated with the risk of multiple diseases and inflammation, it is plausible to think that increased hypocalcemia in over-conditioned cows might be the consequence of reduced appetite or the result of the inflammatory response, as discussed later in this paper.

In dairy cows, excessive body condition increases the risk of fatty liver, which alters hepatic protein synthesis [51]. One possibility is that cows with an increased degree of fatness prepartum might have impaired synthesis of vitamin D binding protein, thus reducing the megalin-mediated endocytosis and, consequently, the supply of precursor for subsequent hydroxylation to 1,25-dihydroxyvitamin D_3_. Vitamin D binding protein is mostly synthesized in the liver [52], and it can bind to all vitamin D metabolites. The vitamin D metabolites 25-hydroxyvitamin D_3_ and the active form 1,25-dihydroxyvitamin D_3_, the latter considered a steroid hormone, can cross the plasma membrane of cells, but they can also enter cells through a receptor-mediated endocytosis. Nykjaer et al. [53] demonstrated that 25-hydroxyvitamin D_3_ bound to the vitamin D binding protein is filtered by the glomerulus and reabsorbed in the proximal convoluted tubules through megalin-mediated endocytosis delivering the cell the precursor for hydroxylation to 1,25-dihydroxyvitamin D_3_ or for catabolism by hydroxylation by the 24-hydroxylase, an enzyme encoded by the *CYP24A1* gene, which results in production of 24,25-dihydroxyvitamin D, an inactive form of vitamin D.

## 3. Gastrointestinal Absorption and Homeostatic Mechanisms Maintaining Blood Calcium

It is well established that gastrointestinal absorption of Ca in monogastrics takes place in the small intestine, but the site of absorption of Ca in ruminants is less well characterized, especially in bovine. Quantitative data summarized by Schröder and Breves [54] showed that the site of Ca absorption is influenced by the amount of Ca consumed by bovine. When Ca intake was limited, amounts less than 100 g/d, the pre-duodenal net Ca absorption was close to zero. On the other hand, when Ca intake was greater than approximately 100 g/d, then pre-duodenal net Ca absorption was greater than zero. These findings probably explain why some experiments showed that most Ca absorption took place pre-duodenum in dairy cattle [55], with little Ca being absorbed in the small intestine. Schröder and Breves [54] observed that as Ca intake increased, the contribution from the pre-duodenum to absorption increased, and pre-duodenal absorption of Ca influenced intestinal absorption. Experiments in which pre-duodenal net Ca absorption was positive resulted in limited to no intestinal net Ca absorption [54], perhaps because pre-duodenal Ca absorption increases the concentrations of Ca^2+^ in blood, with the latter inhibiting the secretion of PTH and thus, synthesis of 1,25-dihydroxyvitamin D_3_ needed to stimulate active Ca absorption in the intestine. However, it is not uncommon for dairy cattle diets to be supplemented with CaCO_3_, which is mostly insoluble at a pH close to neutral or slightly acidic, such as the pH of the rumen fluid. The low solubility of CaCO_3_ would likely limit the amount of Ca^2+^ present in the rumen fluid for absorption pre-duodenum.

Calcium absorption in the gastrointestinal tract occurs through two main pathways: paracellular and transcellular transport (Figure 1A,B). Paracellular transport involves the movement of Ca between adjacent epithelial cells and is energy independent [56]. Epithelial cells form intercellular junctions in their lateral membrane to maintain cellular attachment and tissue integrity [57]. Among those, tight junctions are the major determinant of paracellular permeability and these are formed by a complex of proteins. Within tight junctions, claudin and occludin proteins form pores of different sizes and charge selectivity, which can control the permeability to Ca^2+^ [58]. Claudin-2 and claudin-12 are involved in divalent cation selectivity, such as Ca^2+^ [59]. Although paracellular transport of Ca was previously thought to be an unregulated process controlled solely by chemical gradient, recent evidence suggests a vitamin D-dependent regulatory mechanism. Feeding low Ca diets to goats increased expression of claudin-2 and claudin-12 in the small intestine [60], potentially regulated by increased 1,25-dihydroxyvitamin D_3_ through the classical VDR pathway as it has been shown in mice [59]. Furthermore, nongenomic actions of 1,25-dihyxroxyvitamin D_3_, through membrane-associated rapid-response steroid-binding protein, also resulted in increased paracellular transport of Ca in the intestine of rats [61]. Interestingly, 1,25-dihyxroxyvitamin D_3_ suppresses the expression of the cell adhesion protein, cadherin-17, and tight junction channel aquaporin-8, suggesting that vitamin D regulates the integrity and permeability of epithelial cells that can favor paracellular transport of Ca [62]. In the bovine rumen epithelium, paracellular transport of Ca is supported by in vitro [63] and in vivo studies [64], but the presence of claudin-2 and claudin-12, and their vitamin D-dependent regulation warrants further research. An increased concentration of Ca^2+^ is required in the lumen of the gut to form a positive chemical gradient that favors paracellular transport. Based on the Nernst equation, which relates the numerical value of the concentration gradient to the electrical gradient that balances it, the concentration of Ca^2+^ should be greater than 6 mM in the lumen of the gastrointestinal tract for paracellular transport of Ca to take place [65]. Nevertheless, recent results with bovine rumen epithelia mounted in Ussing chambers showed that increasing the Ca^2+^ concentrations in the luminal side of the epithelium from 1.20 mM to 3.74 or 11.2 mM resulted in a large increase in the net flux of Ca from the mucosa to serosa at the same time that the secretory component of Ca, from serosa to mucosa, remained unaffected [63]. These results suggest a stimulatory effect of Ca^2+^ concentration on Ca absorption in the bovine rumen at concentrations smaller than that theoretically needed to stimulate paracellular absorption. Therefore, the amount of Ca intake, solubility of Ca in different dietary sources, and the gastrointestinal sojourn time are important determinants of paracellular transport of Ca.

Transcellular transport of Ca constitutes the movement of Ca through the epithelial cell and it can be subdivided into three steps. Calcium enters the cell through the apical membrane; it is transported across the cytosol, and Ca exits the cell through the basolateral membrane to the extracellular fluid and blood (Figure 1). In contrast with the paracellular transport of Ca, transcellular transport is an energy-dependent process that does not require a positive concentration gradient. Moreover, transcellular transport is tightly regulated with extensive evidence for hormonal regulatory mechanisms. Uptake of Ca from the lumen into the enterocyte is mediated by the transient receptor potential vanilloid channel (TRPV) subfamilies 5 and 6, in which TRPV6 is only expressed in the intestine [66], whereas TRPV5 is also expressed in the kidney [67]. Because of the low concentration of cytosolic Ca^2+^, approximately 100 nM, and the negative electrical potential inside the cell, Ca transport through the TRPV5 or TRPV6 across the apical membrane is favored by the electrochemical gradient of Ca and does not require energy. Once Ca enters the epithelial cell, it binds to calbindin-D_9k_ in the intestine [68] or calbindin-D_28k_ in the kidney [69]. These proteins have EF-hands region that binds with high affinity to Ca [70], and the complex is translocated through the cell to the basolateral membrane. Nevertheless, Ca movement from the cytosol to the extracellular space is against the concentration and electrical gradient and, therefore, requires ATP. Indeed, the extrusion of Ca from the cell is an energy-dependent process [71], which is reduced by the ATPase inhibitor trifluoroperizine [72]. At the basolateral membrane, Ca is extruded from the cell by the plasma membrane Ca ATPase (PMCA) and, to a lesser extent, by the plasma membrane Na/Ca exchanger 1 (NCX1) [73]. Although NCX1 does not require ATP directly, Na/K ATPase uses ATP to regulate the cytosolic concentration of Na [74] and, therefore, both transporters are energy dependent (Figure 1).

Gastrointestinal Ca absorption and homeostasis are highly regulated by calciotropic hormones such as PTH and vitamin D metabolites. A decline in blood concentrations of Ca^2+^ is sensed by Ca-sensing receptors in the Chief cells of the parathyroid gland [75], which leads to PTH secretion into the blood. Degradation of PTH decreases in hypocalcemia and, therefore, more PTH is available for secretion [76]. In the kidney, PTH increases Ca reabsorption by directly stimulating the mRNA and protein expression of TRPV5. The increased transepithelial Ca^2+^ transport results in increased protein expression of other Ca transport proteins, such as calbindin-D_28k_ and NCX1 [77]. Concurrently, PTH inhibits P uptake in the proximal convoluted tubule by reducing the Na^+^/inorganic phosphate cotransport proteins NaPi-2a and NaPi-2c [78]. Hence, PTH increases Ca and decreases P reabsorption from the urinary filtrate in an attempt to conserve Ca and excrete P.

In the bone, osteoblasts and osteocytes regulate bone remodeling by influencing the expression of the receptor activator of nuclear factor-kappa B ligand (RANKL) and osteoprotegerin (OPG) [79]. The RANKL stimulates bone resorption, whereas OPG prevents the interaction between RANKL to its receptor and stimulates bone formation [79]. Under PTH stimulation, osteoblasts increase synthesis of RANKL which binds to RANK in osteoclast precursors, stimulating cell maturation. In addition, PTH decreases OPG synthesis [80], favoring bone resorption and release of Ca and P into the circulation (Figure 2F). Although the classical target tissues are bone and kidney, PTH also stimulates adenylate cyclase activity in enterocytes [81], which increases the duodenal uptake of Ca [82]. Even though PTH has a direct effect on tissues, resulting in increases in blood Ca, PTH also influences the vitamin D pathway (Figure 2D), which promotes increases in blood Ca by stimulating gastrointestinal absorption (Figure 1A,B).

Synthesis of 1,25-dihydroxyvitamin D_3_ starts by cleavage of the beta ring of 7-dehydrocholesterol, an intermediate in the pathway of cholesterol synthesis, in the skin of animals exposed to ultraviolet-B light radiation forming the secosteroid previtamin D_3_ [83]. In a temperature-dependent reaction [84], previtamin D_3_ undergoes isomerization into vitamin D_3_, which is released into the bloodstream. Vitamin D_3_ can also be absorbed in the small intestine from dietary sources of animal origin, which for bovine would be of minimum contribution. Fungi and yeast present in plants can use ergosterol to form a similar compound, vitamin D_2_ or ergocalciferol, after exposure to UV light. Both vitamins D_3_ and D_2_ can be activated through a series of hydroxylation reactions, first in the liver [85] and then in the kidney [86], to form 1,25-dihydroxyvitamin D. Despite being very similar, vitamin D_3_ is more efficiently converted into 1,25-dihydroxyvitamin D than vitamin D_2_ [87], and data in dairy cattle show that supplementing vitamin D_2_ interferes with the ability of vitamin D_3_ to increase plasma concentrations of 25-hydroxyvitamin D_3_ [88]. In the hepatic mitochondria and microsomes, vitamin D_3_ is converted into 25-hydroxyvitamin D_3_ [89], the major circulating form of vitamin D_3_ metabolites. Under current feeding practices, plasma concentrations in dairy cows range mostly between 40 and 90 ng/mL [90]. Ultimately, 25-hydroxyvitamin D_3_ is transported into the renal mitochondria for hydroxylation of C_1_ by the enzyme 1α-hydroxylase [86], forming 1,25-dihydroxyvitamin D_3_. This enzyme is the target of regulatory hormones and growth factors as it is the rate-limiting enzyme to produce 1,25-dihydroxyvitamin D_3_, thereby being highly regulated by calciotropic hormones [91].

Recent experiments with dairy cows showed that feeding large quantities of vitamin D_3_ (cholecalciferol) did not markedly increase plasma concentrations of 25-dihydroxyvitamin D_3_ [3,92], thereby suggesting that the hepatic 25-hydroxylase likely is under some control. The concentrations of 25-hydroxyvitamin D_3_ in the plasma of individual cows fed 3 mg of vitamin D_3_ for the last 3 weeks of gestation [3] or for a period of 56 d during lactation [92] did not exceed 90 ng/mL. However, when prepartum and mid-lactation cows were fed 1 to 4 mg/d of calcidiol, concentrations of 25-hydroxyvitamin D_3_ exceeded 140 ng/mL [3,92], suggesting that regulatory mechanisms are in place to prevent excessive conversion of vitamin D_3_ into 25-hydroxyvitamin D_3_, which are obviously evaded once cows are fed 25-hydroxyvitamin D_3_. Also, it is clear that one can achieve adequate concentrations of 25-hydroxyvitamin D_3_ in the plasma of cattle with a much smaller amount of calcidiol than cholecalciferol supplied in the diet.

Little is known about the regulation of the 25-hydroxylase enzyme in bovine liver. Two forms of the enzyme have been identified, a mitochondrial and a microsomal enzyme, and serum concentrations of 25-hydroxyvitamin D_3_ are associated with mitochondrial 25-hydroxylase activity in the liver of rats [93]. In cattle, mutations in the *CYP2J2* gene are associated with serum concentrations of 25-hydroxyvitamin D_3_ [94], but 25-hydroxylase activity is often considered an unregulated step in the vitamin D pathway. It is possible that bovine 25-hydroxylase activity is saturated by a large supply of vitamin D_3_ or that other compounds of vitamin D metabolism control expression and activity of 25-hydroxylase such that the conversion of vitamin D_3_ into 25-hydroxyvitamin D_3_ is inhibited. Some evidence suggests that 25-hydroxylase activity is influenced by 1,25-dihydroxyvitamin D_3_ [95], and this effect might be mediated by cytosolic concentrations of Ca^2+^ [95,96].

Regulation of Ca homeostasis by 1,25-dihydroxyvitamin D_3_ requires the latter to bind to VDR. Once 1,25-dihydroxyvitamin D_3_ binds to the VDR, the VDR forms a heterodimer with the retinoid-X receptor (RXR). The complex VDR-RXR heterodimer recognizes vitamin D-responsive elements, which are specific DNA sequences [97]. Localization of the VDR-RXR complex to accessible vitamin D-responsive elements in the genome facilitates induction or repression of transcription as a result of the recruitment of co-activators and co-repressors. When 1,25-dihydroxyvitamin D_3_ binds to VDR, there is disruption of co-repressors that are bound to the VDR, which are replaced by co-activators, resulting in local chromatin relaxation and gene activation [98]. In monogastrics, 1,25-dihydroxyvitamin D_3_ regulates gastrointestinal absorption of Ca mostly in the duodenum [99] by enhancing the expression of the apical membrane transporter, TRPV6 [100], and the basolateral transporters, PMCA and NCX1 [101,102]. Likewise, calbindin-D_9k_ encoded by the gene *S100G* had mRNA expression reduced in the intestines of mice lacking VDR [103]. In ruminants, expression of genes for *VDR*, *S100G*, and *ATP2B1* (gene encoding PMCA1) has been reported in the duodenum of dairy cows [104]. On the other hand, transcriptomic data from RNA sequencing or PCR from six published studies, four using bovine [63,105,106,107] and two using ovine [108,109] rumen tissue showed either absent or marginal abundance of mRNA for *TRPV5*, *TRPV6*, *S100G*, *SLC81A* (gene encoding NCX1), *ATP2B1*, suggesting that the classic mechanisms for Ca absorption and transport across the rumen epithelium are either absent or in limited abundance (Figure 1A). Schröder and Breves [54] showed that the site of absorption of Ca in the bovine gastrointestinal tract is dependent on the amount of Ca consumed. In vitro experiments with Ussing chambers clearly demonstrated the presence of the transcellular transport of Ca^2+^ in the bovine ruminal epithelia in the absence of *TRPV6* expression [63]. Either distinct Ca channels and transport mechanisms exist in bovine rumen epithelia to actively absorb Ca, or the mechanisms are dependent on paracellular transport and the proteins involved, and their importance likely differs from those characterized in the small intestine (Figure 1B). Recently, *TRPV3* has been shown to be expressed in the bovine rumen epithelium with a potential role in the transport of cations, including Ca^2+^ [110]. Thus, the regulatory effects of 1,25-dihydroxyvitamin D_3_ and the cellular mechanism of ruminal transcellular Ca^2+^ transport remain undefined but are important to uptake and transport Ca across the epithelia (Figure 1A).

Vitamin D-dependent increases in Ca availability go beyond the gastrointestinal tract flux of Ca^2+^. The steroid 1,25-dihydroxyvitamin D_3_ stimulates the expression of the RANKL gene (*TNFSF11*) in osteoblasts through vitamin D-responsive elements (Figure 2F), favoring osteoclastogenesis [111]. In addition, mRNA for OPG (*TNFRSF11B*) degradation is accelerated, and *TNFRSF11B* is transrepressed by 1,25-dihydroxyvitamin D_3_ [112]. At the bone matrix, osteoclasts secrete lysosomal enzymes such as cathepsin K to digest the collagen in the organic matrix and release mineral compounds. An acidic media is required to convert Ca salts into soluble forms; thus, hydroxyapatite is dissolved and releases Ca^2+^ and PO_4_^3−^, which can be absorbed by the osteoclast and then released into the circulation. Carbonic anhydrase II, present in osteoclasts, increases the availability of protons transported to the bone matrix, which reduces the local pH. De novo production of carbonic anhydrase II is induced in promyelocytes by 1,25-dihydroxyvitamin D_3_ at the mRNA and protein levels [113], further supporting the vitamin D-dependent bone resorption mechanisms (Figure 2F).

Renal reabsorption of Ca mostly occurs by passive transport in the proximal convoluted tubule and the thick ascending limb of Henle Loop, whereas transcellular reabsorption takes place at the distal convoluted and connecting tubules [114,115]. The kidney excretes only 1 to 2% of filtered Ca, and Ca reabsorption from the filtrate is regulated at the transcellular level, and 1,25-dihydroxyvitamin D_3_ decreases urinary Ca loss, whereas tubular aciduria reduces Ca reabsorption and increases urinary losses (Figure 2D). Presumably, the increased calciuria during renal tubular acidosis involves reduced conductance resulting in closure of the TRPV5 channel [116]. The low pH in the urinary filtrate is sensed by an amino acid residue that serves as a pH sensor located on the surface of the TRPV5 channel [116]. In vitamin D-deficient rats, 1,25-dihydroxyvitamin D_3_ increased the expression of mRNA and protein for TRPV5 in epithelial cells of the distal convoluted tubule and connecting tubule [117]. Moreover, 1,25-dihydroxyvitamin D_3_ enhanced calbindin-D_28K_ expression and accelerated PTH-dependent Ca transport [118], resulting in increased Ca availability in blood.

## 4. Inflammation and Redistribution of Minerals

During the postpartum period, dairy cows are at an increased risk of developing diseases that often induce inflammation, and excessive inflammatory responses might also increase the risk for diseases. Parturition causes trauma and damages uterine and pelvic tissues, favoring bacterial colonization and the development of uterine diseases [119]. Gram-negative bacteria predominate in the uterus of dairy cows in the first days after calving, and concentrations of lipopolysaccharides (LPS) can be high in the uterine fluid of those with metritis [119]. Also, the risk of mastitis increases in the first weeks postpartum [120], and gram-negative bacteria represent a large portion of the isolates causing mastitis in dairy cows.

Gram-negative bacteria that affect the uterus or mammary gland release LPS during the lysis of the cell wall. Lipopolysaccharides are composed of lipid A, a core oligosaccharide, and an O antigen [121], and lipid A is the main pathogen-associated molecular pattern that is recognized by mammalian cells through the Toll-like receptor 4 [122]. Upon recognition of LPS, immune cells activate the NF-κB pathway that leads to the synthesis and release of proinflammatory mediators, including tumor necrosis factor-α, interleukin-6, and interleukin-8 [123]. In dairy cows, intravenous administration of LPS induces systemic inflammation, and one of the many consequences of this acute inflammation is a reduction in the concentrations of Ca and P in serum, while no effect was observed in concentrations of Mg in serum [124]. In agreement, Kvidera et al. [125] showed that the concentration of Ca^2+^ in blood decreased by 46% after LPS administration in dairy cows. In dogs and goats, hypocalcemia has also been observed after an LPS challenge [126,127]. Horses challenged with LPS had hypocalcemia with increased release of PTH that decreased urinary fractional excretion of Ca and Mg but increased urinary loss of P [128], the latter response possibly as an attempt to reestablish Ca balance. There are multiple possible mechanisms whereby inflammation might contribute to hypocalcemia, among them activation of immune cells, impaired vascular permeability, reduced gastrointestinal absorption of Ca, or increased urinary loss of Ca. Nevertheless, under spontaneous disease, such as mastitis caused by gram negative bacteria, the change in blood concentrations of Ca seemed mediated by changes in protein-bound Ca but not Ca^2+^ [129]. Perhaps, the acute changes in Ca^2+^ induced by LPS, as shown by Kvidera et al. [125], might better represent the findings of a very acute disease or sepsis.

Immune cells require Ca^2+^ to be activated. Once stimulated, immune cells increase the uptake of Ca from blood and migrate into the affected tissue. Bovine neutrophils during the resting state have a cytosolic concentration of Ca^2+^ of 85 nM, which increases to 300 to 400 nM upon activation [130]. Although there is a four-fold increase in the immune cell cytosolic Ca^2+^ concentration, not all the Ca^2+^ originates from blood, as some is released from the endoplasmic reticulum [131], and calculations of Ca^2+^ uptake by immune cells suggest only a minor role in the changes in blood Ca concentrations with the purpose of activating leukocytes. For instance, considering the volume of neutrophils of 1766 μm^3^, 50% being cytosol, an increase in cytosolic Ca^2+^ of 315 nM, and 50% of the leukocytes in the entire blood pool experiencing activation, would result in uptake of only 0.03 ng of Ca per mL of blood, which represents only 0.00007% of the Ca^2+^ concentration in blood of 1.2 mM. Therefore, using this simplistic static approach to calculate potential changes in blood Ca^2+^, it is unlikely that uptake of Ca^2+^ from the extracellular space to reestablish cytosolic concentrations during immune activation plays any role in the drop in blood Ca concentrations during inflammation.

Another mechanism involved in inflammation-induced hypocalcemia is the altered vascular permeability caused by the inflammatory process. Neutrophils migrate from blood to the site of infection to phagocytize and kill pathogens [132]. Thus, neutrophils interact with the endothelium at the target tissue through adhesion molecules and secrete proteases, elastase, and cathepsin G to destroy the integrity of endothelial cells and basement membrane [133]. In addition, neutrophils secrete matrix metalloproteinases [134] that help degrade the basal membrane and interstitial tissue proteins to facilitate the process of migration. Consequently, vascular permeability increases and neutrophils can reach the site of infection. This adjustment of the endothelial barrier enables blood constituents to exit the circulation and enter the extravascular tissues. Albumin is the most abundant protein in blood and leaks from the blood pool during endotoxemia. Because it is negatively charged, albumin normally binds Ca and protons. Approximately 40 to 45% of the blood pool of Ca is bound to albumin; however, during alkalosis, there is a reduction in the concentration of protons, and more Ca^2+^ binds to albumin, leading to a reduction in the proportion of Ca^2+^ (Figure 2A). Administration of LPS to cows has been shown to induce alkalosis [135], which could contribute to the reduction in Ca^2+^ observed in LPS-treated cows [125]. In fact, cows administered LPS intramammary had the appearance of bovine serum albumin and trypsin inhibitor capacity in the whey of milk [136]. Furthermore, intravenous administration of *Escherichia coli* toxin O111:B4 to pigs resulted in hypocalcemia and increased the peritoneal fluid concentration of Ca from 1.35 to 2.10 mM [137], supposedly induced by increased vascular permeability as a large increase in Ca was detected in the peritoneal fluid and liver.

A third mechanism potentially involved in inflammation-induced hypocalcemia is the reduction of Ca absorption from the gastrointestinal tract. Lipopolysaccharides reduce gastrointestinal motility in in vivo and in vitro studies [138,139], resulting in increased gastric retention in a dose-dependent manner [140]. The effects of LPS on intestinal contractibility seem to be partially mediated by direct binding to the Toll-like receptor 4 [141] and by promoting the synthesis of proinflammatory molecules by immune cells, such as tumor necrosis factor-α and interleukin-1 [142]. In dairy cows, LPS administration reduced reticulo-ruminal contractions [143]. In agreement, in small ruminants, LPS administration reduced muscular contractility in the rumen, reticulum, abomasum, and duodenum [144]. Motility of the gastrointestinal tract is important to allow contact of minerals to the apical membrane of epithelial cells and facilitate absorption. Reduced gut contractility would likely impair the absorption of Ca, contributing to the hypocalcemia observed in animals receiving LPS. Furthermore, administration of LPS to mice decreased intestinal mRNA expression of *TRPV6*, *S100G*, and *SLC34A2* (gene encoding NaPi-2b), thereby suggesting potential perturbations in intestinal Ca and P transport [145]. Nevertheless, further research is warranted to determine if systemic inflammation alters the gastrointestinal absorption of Ca.

Finally, it is possible that endotoxins impair Ca reabsorption from the renal filtrate, resulting in increased calciuria. However, administration of LPS to cows can induce alkalosis [135], and alkaline urine pH favors the action of the TRPV5 to reabsorb Ca from the urinary filtrate [116,146]. In mice, administration of LPS induced hypocalcemia and hyperphosphatemia, resulting in increased PTH and consequent renal expression of *CYP27b1* and increased 1,25-dihydroxyvitamin D_3_ [145]. Endotoxemia caused mixed effects on renal mRNA and protein expression for Ca and P channels and binding proteins. The mRNA and protein expression of channels TRPV5 and TRPV6 were both upregulated, whereas protein expression for calbindin-D_28k_ and NCX1, and the mRNA for NaPi-2a (*SLC34A1*) and NaPi-2c (*SLC34A3*) were downregulated by LPS, showing marked effects on the FGR-23/vitamin D axis. Despite the alterations in renal transport proteins for Ca and P, authors did not find increased calciuria or phosphaturia in mice treated with LPS. Moreover, urinary losses of Ca and P decreased following LPS challenge [145].

Collectively, if mechanistic models apply interspecies, it is likely hypocalcemia, often observed following an inflammatory insult, is caused primarily by the redistribution of Ca away from the vasculature. Gastrointestinal absorption might be altered either by stasis or reduced transport across the gastrointestinal mucosa. Therefore, inflammation that often occurs during or after parturition likely contributes to the increased risk of hypocalcemia in early postpartum cows.

## 5. Mechanisms Linking Hypocalcemia and Health in Dairy Cows

Hypocalcemia has been associated with several periparturient disorders (Figure 3), including calving-related problems such as dystocia, retained placenta, and uterine prolapse [147,148,149]; metabolic disorders such as hyperketonemia and displaced abomasum [147,150]; and uterine diseases such as metritis [16]. Physiological processes that influence these outcomes include decreased smooth muscle function [151], which results in the reduced rumen and gastrointestinal motility [14,152] and uterine motility [153]. Induction of hypocalcemia directly decreases DMI [14], and prevention of hypocalcemia by dietary means prepartum increases DMI postpartum [44,154], and cows that develop milk fever have impaired immune response [13]. Furthermore, Martinez et al. [14] showed that induced hypocalcemia in dry cows reduced concentrations of cytosolic Ca^2+^ and compromised phagocytosis and oxidative burst of neutrophils. Interestingly, the repletion of extracellular Ca in the media of neutrophils derived from hypocalcemic cows improved their phagocytic activity [155]. It is important to mention that spontaneous subclinical hypocalcemia have different presentations, and cows classified as having transient hypocalcemia, primiparous with plasma total Ca ≤ 2.15 mM on the day following calving and Ca > 2.15 mM at 2 days postpartum, and multiparous cows with plasma total Ca ≤ 1.77 mM on the day after calving and Ca > 2.20 mM on day 4 postpartum, produced the most milk in the first 10 weeks postpartum and had similar risk of diseases compared with cows classified as normocalcemic in the first days postpartum [11].

Calcium signaling is critical for the function of neutrophils, which are important cells in defense against infections of the uterus and mammary gland. Neutrophils are among the first immune cells to migrate from circulation to the site of infection. They are stimulated by the engagement of ligands to G-coupled receptors and Fcy-receptors, activation of β2-integrins, or the interaction of E-selectin with P-selectin glycoprotein ligand 1 and L-selectin. This stimulus leads to the hydrolysis of phospholipids by phospholipase C forming diacylglycerol and inositol-1,4,5 triphosphate, which will promote the release of Ca^2+^ stored within the endoplasmic reticulum [163]. The reduction of Ca^2+^ within the endoplasmic reticulum needs to be replenished; therefore, activation of Ca ATPase pumps within the endoplasmic reticulum allows for entry of Ca^2+^ from the cytosol to fulfill the replenishment of Ca [164]. Concurrently, replenishment of cytosolic Ca is accomplished by activation of plasma membrane Ca channels, thus allowing Ca to be taken up by the cell from the extracellular space, a process, namely store-operated Ca entry [165]. This increased influx of Ca into the cell controls several cellular functions, including cell proliferation, enzymatic activity, and death [166]. Within the neutrophil phagosome, reactive oxygen species are produced from superoxide anions and are used for microbial killing. A key enzyme in the synthesis of reactive oxygen species and the development of the inflammatory process is the NADPH oxidase, which is regulated by the influx of Ca^2+^ [167]. Inadequate concentrations of blood Ca^2+^ are expected to impair the proper entry of Ca^2+^ into neutrophils through the store-operated Ca entry mechanism, compromising cell organelle Ca replenishment and cell function, the latter observed in dairy cows induced to have subclinical hypocalcemia [14]. Ultimately, hypocalcemia reduces the ability of cows to fight bacterial infections and may increase the susceptibility to uterine and mammary gland infection [12]. Calcium can also influence immune function by affecting the synthesis and secretion of cytokines. Gene expression for interleukin-8 (*CXCL8*), which is an important chemokine that activates and recruits neutrophils into the site of infection, depends on the elevation of cytosolic Ca^2+^ concentration [168].

Calcium signaling contributes to the production of cytokines that signal among immune cells. Cytokines categorized as proinflammatory (e.g., interleukin-1 and tumor necrosis factor-α) promote systemic inflammation, whereas those anti-inflammatory (e.g., interleukin-4 and interleukin-10) serve as a feedback to limit the proinflammatory response [169]. Increased concentrations of Ca^2+^ within the cytosol activate calcineurin, a serine/threonine protein phosphatase, by binding Ca^2+^ to the Ca-binding regulatory subunit of calcineurin [170]. The nuclear factor of activated T cells (NFAT) is a transcription factor that regulates gene expression and synthesis of several cytokines, but when phosphorylated, it cannot enter the nucleus. Thus, dephosphorylation of NFAT by calcineurin is required for nuclear translocation and also to increase the affinity of NFAT to specific DNA locations [171]. Because NFAT proteins are rapidly exported from the nucleus, calcineurin activity needs to be maintained by a persistently high concentration of Ca^2+^ in the cytosol [172]. The role of NFAT in regulating the immune and inflammatory response is wide and can vary depending on cell type. The calcineurin/NFAT signaling pathway is present in most immune cells, including neutrophils, eosinophils, basophils, macrophages, natural killer cells, and dendritic cells and, depending on the cell synthesis of proinflammatory or anti-inflammatory mediators, can be induced by NFAT [173]. Thus, the balance of the immune system is tightly regulated to prevent the exacerbation of inflammatory responses. Hypocalcemia can affect cytosolic concentrations of Ca^2+^ [14], which potentially alters the calcineurin/NFAT signaling, unbalancing the immune response and contributing to increased susceptibility to inflammatory processes and infectious diseases.

Collectively, there is strong evidence supporting the role of Ca in optimal immune function. Impairment of immune response occurs simultaneously with decreased muscle contractility in hypocalcemic cows. Thus, it is not surprising that hypocalcemia is considered a gateway disease [31] and that meta-analyses [44,154] identified reduced risks of diseases with prevention of hypocalcemia by reducing the dietary cation-anion difference (DCAD) of prepartum diets.

## 6. Methods to Prevent Periparturient Mineral Imbalance

### 6.1. Alterations in Acid-Base Balance

Feeding acidogenic diets to prepartum dairy cows is a common method to improve Ca homeostasis and reduce the risk of hypocalcemia. The discovery of the relationship between cation and anion balance and risk of hypocalcemia in dairy cows in the late 50s and 60s was initially random, after the addition of organic acids to improve forage preservation during ensiling. These forages were fed to dry cows, and investigators later observed increased blood Ca and reduced incidence of clinical hypocalcemia [174]. Later, Ender et al. [175] suggested that the DCAD could be calculated by the following equation: DCAD = ([mEq of K^+^ + mEq of Na^+^] − [mEq of Cl^−^ + mEq of S^2−^]), which still is the most commonly used equation to calculate cation-anion difference in diets by dairy nutritionists. The addition of other cations, such as Ca^2+^ and Mg^2+^, and other anions, such as PO_4_^3−^, to the equation has been suggested; however, there is no clear evident benefit of adding those ions with their respective bioavailabilities to calculate DCAD and predict the risk of hypocalcemia in dairy cows [36,176,177]. Block [178] brought the concept to North America and further investigated the use of acidogenic diets for prepartum cows. His findings showed that not only was hypocalcemia reduced, but milk production increased by reducing the DCAD of the prepartum diet. Two recent meta-analyses of the literature demonstrated that reducing the DCAD of prepartum diets reduced the risk of hypocalcemia and the number of disease events per cow, mostly because of a reduction in milk fever, retained fetal membrane, and metritis [44,154]. Their findings clearly showed that acidogenic diets benefit multiparous cows by increasing postpartum intake and yields of milk and fat-corrected milk [44,154]. However, the authors also identified the need to further understand the role of feeding acidogenic diets to nulliparous cows because of the scarcity and heterogeneity of the data currently available. Indeed, recent work showed limited benefits to feeding acidogenic diets to prepartum nulliparous cows [179,180]. Diet-induced metabolic acidosis is achieved by manipulating the mineral composition of the diet to result in a negative DCAD [181]. For instance, feeding acids of strong anions, such as HCl and H_2_SO_4_, or feeding salts containing strong anions, such as CaCl_2_, MgSO_4_, and NH_4_Cl, reduce the DCAD [17]. When those ingredients are fed to cows, the absorption of anions is greater than that of cations, or the cation is metabolized within the gastrointestinal tract and only partially absorbed as such, thereby increasing the absorption of negatively charged ions. The absorption of many anions occurs, in part, in exchange with HCO_3_^−^, resulting in reduced base excess in blood. Also, anion absorption results in the retention of H^+^ to maintain equivalence in cell or body fluid electric charges. This process of loss of HCO_3_^−^ and retention of H^+^ results in a reduction in the base excess and a concurrent reduction in blood pH. Thus, feeding diets with a negative DCAD to cows leads to a state of compensated metabolic acidosis with reduced partial pressure of CO_2_ and increased urinary excretion of H^+^ [2,3,179,181]. If uncompensated, then metabolic acidosis might have detrimental effects on the cow [182,183].

Under metabolic acidosis, Ca homeostasis is altered by different mechanisms (Figure 2). A drop in blood pH alters the balance of total Ca to Ca^2+^, favoring ionization of Ca by releasing it from bound to albumin and complexed with salts such as bicarbonate, lactate, phosphate, or citrate in the blood (Figure 2A). Also, during metabolic acidosis, the concentration of bicarbonate in the blood is reduced; therefore, Ca–bicarbonate interactions are reduced, resulting in the release of Ca^2+^ [184]. Consequently, the change of Ca^2+^ in blood during metabolic acidosis is greater than during respiratory acidosis [185]. In fact, in vitro manipulations of blood pH resulted in a rate of increase of 0.36 mM in Ca^2+^ concentrations with the reduction of one pH unit [186], although such a drastic change in pH should not occur in blood because it threatens life.

Induction of metabolic acidosis increases the secretion rate of PTH [185] (Figure 2B), enhances tissue responsiveness to PTH with increases in blood Ca and 1,25 dihydroxyvitamin D_3_ [187,188] (Figure 2C), and increases the expression of PTH receptors (*PTH1R*) in the renal cortex [189] (Figure 2D). These collective effects of metabolic acidosis on calciotropic hormones likely contribute to the reduced risk of hypocalcemia postpartum. In addition to the direct effects of PTH and 1,25-dihydroxyvitamin D_3_ on bone resorption (Figure 2F), the increased concentration of H^+^ during metabolic acidosis directly increases the net flux of Ca from bone calvariae [190]. Osteoblasts produce prostaglandin E_2_ during acidosis, which stimulates the synthesis of RANKL-promoting bone resorption [191]. The inhibition of prostaglandin E_2_ synthesis limited bone Ca release and RANKL expression induced by acidosis [192]. Thus, metabolic acidosis directly increases the release of Ca from the bone reservoir by affecting osteoclast activity, which is mediated by prostaglandin E_2_. Bone is a large reservoir of buffers, and increased bone resorption facilitates blood buffering during metabolic acidosis. Thus, acidosis increases mineral efflux from bones initially through physicochemical mechanisms to buffer blood, followed within hours by cell-mediated mechanisms that culminate with increased osteoclastic activity and bone resorption [192].

Feeding diets that result in diet-induced compensated metabolic acidosis has been shown to increase gastrointestinal absorption of Ca in dairy cows [179,188], presumably by increased Ca flux across the rumen epithelium [193]. It is suggested that the changes in the PTH–vitamin D axis previously discussed favor gastrointestinal Ca transport, as depicted in Figure 2E.

Despite the benefits of feeding acidogenic diets to postpartum DMI [44], diet-induced metabolic acidosis is known to decrease prepartum intake [44,154,176]. Such an effect has been demonstrated whether cows are fed acidogenic salts or commercial products [154]. Recent work clearly showed that metabolic acidosis underlies the depression in DMI and alters feeding behavior in dairy cows [6]. When the diet containing the acidogenic product was supplemented with alkalogenic salts to result in the same positive DCAD and acid-base balance as that of cows fed diets without any acidogenic product, then the depression in DMI was prevented [6]. Therefore, the effects of acidogenic diets on DMI are mediated by the changes in the acid-base status of the cow. Depression in intake is one of the reasons to avoid feeding extremely acidogenic diets. Also, acidogenic diets can alter cellular energy metabolism and impair insulin secretion and adipose tissue insulin sensitivity in dairy cows, shifting cellular signals to be less lipogenic and more lipolytic [183]. When cows were fed a diet with −405 mEq/kg of dry matter, measures of energy metabolism were altered with reduced release of insulin in response to a glucose tolerance test [182]. Time series analysis of blood concentrations of Ca, fatty acids, ketones, and glucose from periparturient cows fed an acidogenic diet identified potential feedback mechanisms between Ca and fatty acids [156]. Increased concentrations of fatty acids, glucose, and ketones were observed subsequent to increased Ca concentrations, suggesting a possible role of Ca in the energy metabolism of cattle, a role consistent with that in other species. Nevertheless, it is important to note that the optimum DCAD to control hypocalcemia and improve postpartum performance remains unknown. The available data from experiments with parous cows suggest that there is no need to feed diets with a DCAD smaller than −120 mEq/kg of dry matter to benefit health and production [44].

Limited data exist for how long acidogenic diets should be fed prepartum to elicit its benefits postpartum. Cows develop all hallmark signs of metabolic acidosis within 24 to 36 h following intake of an acidogenic diet, and increased gastrointestinal Ca absorption and improved response to PTH are observed within 3 d of exposure to the diet [188]. Therefore, it is likely that the benefits of those diets in improving Ca homeostasis are to be observed within days. Three experiments evaluated the effects of extending the feeding of acidogenic diets prepartum from the traditional 21 d to either 6 or 8 weeks [2,194,195]. None of the experiments showed any benefit to postpartum performance or metabolism by extending the feeding of acidogenic diets beyond 21 d. In fact, Lopera et al. [2] observed potential negative effects with reduced milk yield and extended days open. Indeed, Lopera et al. [2] pointed out that in three experiments [2,194,195], the yield of energy-corrected milk was always less for cows fed the acidogenic diets longer than 21 d. Two observational studies evaluating days in the prepartum group, in which parous cows were fed acidogenic diets, showed that postpartum performance was optimized when the length of exposure to the diets was 3 to 4 weeks [196,197]. Collectively, the available data indicate that prepartum parous cows should be fed acidogenic diets planned for the last 21 d of gestation.

### 6.2. Low Ca Diets and Sequestering Agents

Limiting the availability of Ca in the gastrointestinal tract successfully reduced the incidence of clinical hypocalcemia [198]. Calcium-restricted diets lead to a small reduction in blood concentration of Ca sufficient to affect the cow’s Ca-sensing receptor “set-point” that causes increased secretion of PTH and synthesis of 1,25-dihydroxyvitamin D_3_ [199]. Consequently, bone resorption is stimulated, as reflected in increased circulating hydroxyproline [200]. One of the challenges is to identify suitable dietary ingredients that have very low concentrations of Ca such that prepartum diets supply less absorbable Ca than required by the cow. A prepartum Holstein cow requires approximately 20 to 30 g/d of absorbable Ca, 9 to 10 g for uterine tissue accretion [4], 10 g to replenish the daily fecal losses when consuming 10 to 11 kg of dry matter daily [5], and another 1.5 to 10 g to meet the urinary losses depending on the type of diet fed [3,6]. Assuming an alkalogenic diet, the prepartum cow would require approximately 22 g/d of absorbable Ca to meet its daily needs. Therefore, to induce negative Ca balance, the diet would have to provide daily no more than 20 g/d of absorbable Ca or up to 35–40 g/d of total Ca, depending on the expected bioavailability [5,201].

Alternatively, gastrointestinal Ca absorption can be reduced using Ca binders, such as products containing aluminum silicates, which would be more practical and potentially mimic the low Ca diet to induce a negative balance prepartum [202]. Zeolites are aluminum silicates with porous structures that can adsorb different cations. Studies in vitro using ruminal fluid demonstrated that zeolite binds Ca and Mg and, under reduced pH, increases the binding of P, thus decreasing the bioavailability of those minerals in dairy cows [203]. Feeding synthetic zeolite for the last 21 d of gestation to dairy cows resulted in greater concentrations of Ca in plasma in the last 5 d before calving, and during the first 3 d postpartum [204]. A common finding when the synthetic zeolite is fed is a reduction in concentrations of P in blood, presumably by reduced gastrointestinal absorption. Also, reducing gastrointestinal Ca bioavailability with zeolite might increase circulating PTH to upregulate Ca absorption mechanisms, which increases urinary excretion of P, thus contributing to the reduced plasma phosphate. Additionally, zeolites may release aluminum, which can decrease the gastrointestinal absorption of P [205]. The reduced blood P concentrations might reduce FGF23 secretion by osteocytes, which would benefit vitamin D-mediated gastrointestinal Ca absorption and bone resorption, although such mechanisms remain to be shown in bovine.

One limitation of zeolites is the amount of ash that they add to the diet. They are typically fed at 5 to 7% of the diet dry matter, which might explain the reduced DMI observed prepartum [204]. Collectively, limiting Ca and P absorption by feeding low Ca and P diets or supplementing sequestering agents are effective in preventing clinical hypocalcemia, but it remains unknown if those strategies benefit postpartum production performance and risk of diseases other than hypocalcemia.

### 6.3. Sources of Vitamin D

According to the NRC [5], all classes of dairy cattle, including prepartum and lactating cows, should receive 30 IU/kg of body weight, which would translate into approximately 20,000 IU or 0.5 mg of vitamin D_3_ daily for a prepartum cow. The new NASEM [201] suggests an adequate intake of 30 IU/kg of BW for dry cows and 40 IU/kg of BW for cows during lactation, although the NASEM dairy software (version 8 R2023.09.15) calculates 32 IU/kg of BW and not 30 for dry cows. The suggested amount is probably the minimum needed to maintain blood concentrations of 25-hydroxyvitamin D_3_ in dairy cows above 30 ng/mL [90]. In many cases, cows are fed amounts above the minimum recommended [90]. Horst et al. [206] suggested that adequate concentrations of 25-hydroxyvitamin D_3_ in cattle range from 20 to 50 ng/mL. Nevertheless, recent data from dairy cows under different feeding regimens showed that most have a concentration of 25-hydroxyvitamin D_3_ between 50 and 80 ng/mL [90]. It is interesting to note that even at very large quantities of vitamin D3 fed to cows, 12,000 IU/d (3 mg of cholecalciferol), corresponding to 4 to 5 times the amounts recommended by the NASEM [201] and NRC [5], respectively, the plasma concentrations of 25-hydroxyvitamin D_3_ in dairy cows did not increase above 100 ng/mL after 3 to 4 weeks of feeding [3,92]. On the other hand, feeding 1 to 3 mg of 25-hydroxyvitamin D_3_ more efficiently increased plasma concentrations of the vitamin in prepartum or lactating dairy cows [3,92]. Supplementing 3 mg/d of 25-hydroxyvitamin D_3_ to prepartum cows fed an acidogenic diet resulted in improvements in Ca homeostasis during the transition period [207]. It is important to note that the benefits of feeding 25-hydroxyvitamin D_3_ on blood Ca^2+^ concentrations were only observed when combined with an acidogenic diet [3]. When supplemented with an alkalogenic diet (+144 mEq/kg), then feeding 3 mg/day of 25-hydroxyvitamin D_3_ prepartum resulted in the smallest blood Ca^2+^ concentrations, particularly in older cows [3]. Furthermore, supplementing 3 mg/d of 25-hydroxyvitamin D_3_ to cows in the last 3 weeks of gestation reduced the incidence of uterine diseases [208], perhaps because of the modulatory effects of vitamin D on the immune system [46]. Cows fed 3 mg/d of 25-hydroxyvitamin D_3_ during the prepartum period had an increased yield of milk in the subsequent lactation [209,210]. In contrast, feeding 6 mg/d of 25-hydroxyvitamin D_3_ showed potentially detrimental effects on cows [211]. Nevertheless, calves born from supplemented cows had increased concentrations of 25-hydroxyvitamin D_3_ in plasma, suggesting a potential benefit of 25-hydroxyvitamin D_3_ supplementation to the dam to improve vitamin D status in calves [211].

The most active form of vitamin D, 1,25-dihydroxyvitamin D_3_, has also been evaluated as preventative to hypocalcemia in dairy cows [212]. Cows starting lactation 3 or greater and fed a diet to increase the risk of hypocalcemia received 500 μg of 1,25-dihydroxyvitamin D_3_ orally and then were grouped according to the day the treatment was administered relative to calving: within 24 h of calving, 1 to 3 d before calving, or 4 to 5 d before calving. Those treated 1 to 3 d before calving had increased blood Ca and P compared with all other groups, including the untreated control [213]. Intramuscular administration of a 1,25-dihydroxyvitamin D_3_ analog to prepartum cows susceptible to developing hypocalcemia, starting 7 d before the expected day of calving and repeating every 7 d until calving, reduced the incidence of clinical hypocalcemia from 85% in untreated controls to 43 and 29% in cows that received 100 and 150 μg of 24-F-1,25-dihydroxyvitamin D_3_; however, the repeated treatments prepartum impaired endogenous 1,25-dihydroxyvitamin D_3_ synthesis postpartum in treated cows and eventually resulted in clinical hypocalcemia [214]. Thus, predicting the day of calving so that vitamin D_3_ metabolites are administered at the proper timing has been a challenge. An alternative is to use 1,25-dihydroxyvitamin D_3_ immediately after calving in an attempt to minimize the risk of hypocalcemia following calving, but the benefits would be observed 12 to 24 h later, which is the time needed for blood concentrations of Ca to increase [1,50]. Subcutaneous administration of 300 μg 1,25-dihydroxyvitamin D_3_ within the first 6 h postpartum improved Ca homeostasis, resulting in increased blood Ca^2+^ and reduced prevalence of subclinical hypocalcemia [1,50]. Additionally, 1,25-dihydroxyvitamin D_3_ improved measures of innate immune function, which might offer benefits to postpartum health [1,50].

### 6.4. Oral Calcium after Calving

Supplementation with oral Ca, either as an oral solution, bolus, or gels containing inorganic or organic salts of Ca, has been extensively used to prevent clinical and subclinical hypocalcemia in dairy cows. The Dairy 2014 survey by the National Animal Health and Monitoring System on dairy cattle management practices in the United States described that 68.9% of the dairy farms surveyed use some oral or injectable Ca as part of their management or for the therapy of cows [215]. Recent experiments have evaluated Ca boluses containing a mixture of chloride and sulfate salts of Ca, likely because boluses are perceived to be safer to administer compared with gels, oral solutions, or intravenous solutions. Feeding approximately 50 g of Ca as CaCl_2_ in solution increased plasma concentrations of total Ca in Jersey cows for at least 6 h [216]. On the other hand, providing 43 or 86 g of Ca as an oral bolus containing chloride and sulfate salts of Ca increased blood Ca^2+^ and serum total Ca for 2 to 6 h, depending on the dose [12]. Despite providing strong anions, the Ca boluses given to postpartum cows did not affect measures of acid-base balance in the hours or days after treatment [12]. Although oral Ca increased blood Ca concentrations transiently, the impacts on health, production, reproduction, and survival have been heterogeneous [217,218]. In some cases, such as supplementation to first lactation cows resulted in negative effects on reproduction [217], or supplementation to multiparous cows increased the risk of culling [218]. A recent meta-analysis of the published literature, including nine experiments with 6670 dairy cows in which oral Ca bolus was used after calving, showed no benefit to milk yield or pregnancy at first postpartum insemination [219]. The lack of benefits from blank intervention with oral Ca supplements at calving likely reflects the fact that some subgroups of cows might benefit from the intervention, whereas other cohorts might suffer detrimental effects [217,218]. If used, oral Ca should target specific populations of cows, those at greater risk for hypocalcemia, such as cows not fed prepartum diets to prevent hypocalcemia, older cows [10], and cows that are more likely to have persistent or delayed hypocalcemia [11] such as those with problems at calving.

## 7. Conclusions

At the onset of lactation, dairy cows undergo perturbations in Ca homeostasis that result in some developing milk fever, and a large proportion undergoes a period of transient subclinical hypocalcemia. Numerous mechanisms are activated to maintain a tight regulation of blood Ca concentrations to avoid hypo or hypercalcemia. Those mechanisms involve the calciotropic hormones PTH and vitamin D. They are responsible for regulating gastrointestinal absorption of Ca, bone remodeling, and the renal excretion of Ca in order to maintain blood Ca between 2.2 and 2.5 mM in most adult cattle. Absorption of Ca from the gastrointestinal tract occurs through two mechanisms: paracellular and transcellular transport. In bovine, the site of Ca absorption seems to depend on intake of the mineral. When intake is limited, then most Ca is absorbed post-abomasum, whereas large intakes result in absorption primarily pre-duodenum. Evidence exists for both paracellular and transcellular mechanisms of Ca absorption to be present in the rumen epithelium; however, the exact pathway and transporters involved in the transport across the rumen wall remain unclear. Hypocalcemia predisposes cows to other periparturient diseases, presumably because of the role of Ca as a second messenger system, but also through inhibition of smooth muscle function. Cows develop hypocalcemia because of irreversible loss of the mineral in colostrum and milk, although inflammation in early lactation can further exacerbate the condition by redistribution of Ca away from the vasculature. Feeding prepartum diets low in Ca and P, which result in negative Ca and P balance, promotes improvement in blood Ca postpartum and reduces the risk of clinical hypocalcemia. Feeding sequestering agents that limit P and Ca absorption prepartum promotes improvements in blood Ca postpartum. Prevention of hypocalcemia by manipulation of the DCAD to induce metabolic acidosis is strongly supported by the literature with sound mechanistic data and extensive experimentation demonstrating improvement in Ca homeostasis, health, and lactation performance postpartum. Additionally, the use of vitamin D metabolites to overcome the challenge of maintaining Ca homeostasis in combination with acidogenic diets may result in additional benefits to animal performance.

## Figures and Tables

**Figure 1 animals-14-01232-f001:**
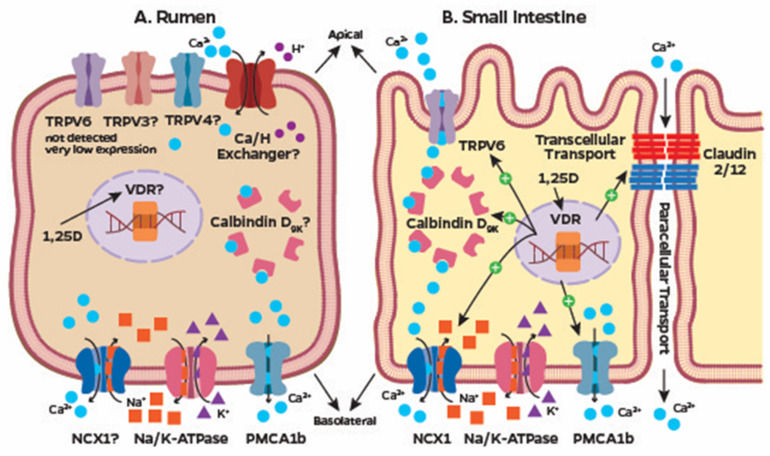
Proposed gastrointestinal absorption of calcium (Ca) in bovine. (**A**), in vitro studies suggest the presence of active transport of Ca in the rumen epithelium; however, it remains unclear the exact mechanism and the Ca channels involved in the apical absorption of Ca. The expression of transient receptor potential vanilloid 6 (TRPV6), the classical channel present in enterocytes, is poorly expressed in rumen epithelium. Alternatively, TRPV3, TRPV4, and a Ca/H^+^ exchanger are potential channels involved in Ca uptake at the apical membrane. The presence of vitamin D receptors and how 1,25-dihydroxyvitamin D_3_ regulates Ca transport in the rumen remains unknown. Likewise, the presence of calbindin-D_9k_ is uncertain. At the basolateral membrane, the ruminal epithelia express plasma membrane Ca ATPase (PMCA1b), which extrudes Ca from the cell; however, the presence of sodium (Na) Ca exchanger 1 (NCX1) needs to be determined. (**B**) On the apical membrane of enterocytes, TRPV6 is primarily responsible for the uptake of Ca into the cell. Once in the cytosol, Ca binds calbindin-D_9k_, and the complex is translocated to the basolateral membrane. On the basolateral membrane, Ca is released from calbindin-D_9k_ and can be extruded from the cell through the NCX1, which is energy independent; however, because it increases the cytosolic concentration of Na, Na/K-ATPase uses ATP to maintain Na balance by exchanging Na with potassium (K), thus maintaining the cell potential. In addition, Ca can be extruded from the cell by the PMCA1b incurring in ATP expenditure. 1,25-dihydroxyvitamin D_3_ plays a key role in Ca transport in the enterocyte by stimulating mRNA and protein expression of TRPV6, calbindin-D_9k_, and NCX1. Calcium absorption also occurs via paracellular transport in favor of the chemical gradient, but recent evidence suggests some control of tight junction proteins claudin 2 an 12 by vitamin D, which can affect the passive flow of Ca from the lumen of the gastrointestinal tract to the interstitial space and venules draining the gastrointestinal tract. Arrows point to the direction of flow of a chemical element, the side of the cell if apical or basolateral, or the effect of a stimulus (hormone or activation of gene) on multiple cellular responses. Arrows with positive symbols indicate downstream stimulation. Different shapes (circles, squares, triangles) with respective colors represent different minerals in and out of the cell. Pocket shapes represent calbindin-D_9k_ protein transporting Ca^2+^ within the cytosol. Channels in the apical and basolateral membranes represent the different ion channels responsible for Ca^2+^ flux in and out of the cell or for maintenance of cell potential.

**Figure 2 animals-14-01232-f002:**
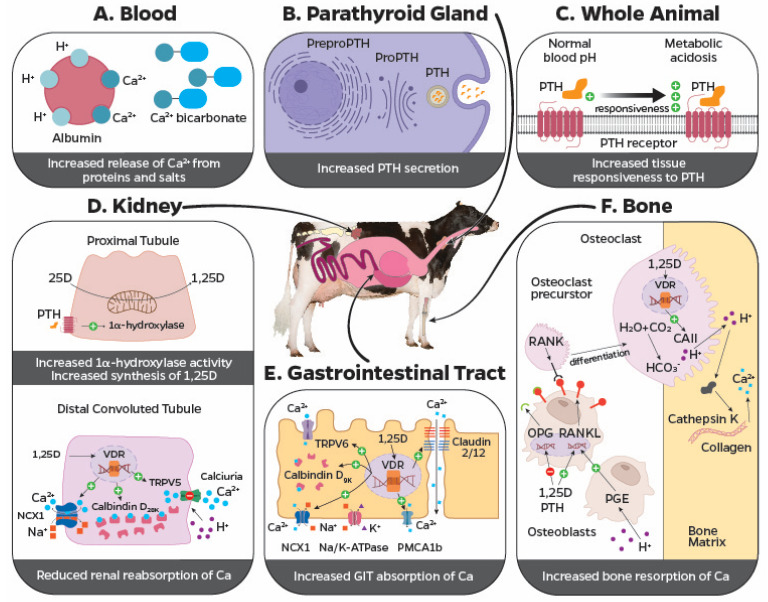
Mechanisms of metabolic acidosis influencing calcium (Ca) homeostasis in bovine. (**A**) Cations such as Ca^2+^ and H^+^ can bind to negatively charged proteins such as albumin. During metabolic acidosis, the increased concentration of H^+^ competes with Ca^2+^ to bind to those proteins, and Ca^2+^ is released. Moreover, Ca binds to anions, such as bicarbonate, and the reduced concentration of bicarbonate during metabolic acidosis results in an increased concentration of Ca^2+^ in the blood. (**B**) In the Chief cells of the parathyroid gland, metabolic acidosis increases the synthesis and release of parathyroid hormone (PTH). (**C**) Tissue responsiveness to PTH is increased during metabolic acidosis; conformational changes in the receptor increase the ability of PTH to bind, thereby increasing the effects of PTH on target cells. (**D**) In the proximal tubule cells of the kidney, PTH stimulates the expression of 1α-hydroxylase in the mitochondria, resulting in greater conversion of 25-hydroxyvitamin D_3_ to 1,25-dihydroxyvitamin D_3_. In the distal convoluted tubule, 1,25-dihydroxyvitamin D_3_ stimulates the expression of transient receptor potential vanilloid 5 (TRPV5), calbindin-D_28k_, sodium Ca exchanger 1 (NCX1), which is expected to increase reabsorption of Ca from the urinary filtrate; however, the resulting tubular acidosis induced by increased concentration of H^+^ in the filtrate blocks the transport of Ca across the TRPV5 resulting in increased urinary loss of Ca. (**E**), absorption of Ca in the gastrointestinal tract (GIT) differs between the rumen and small intestine (see Figure 1). In the small intestine, 1,25-dihydroxyvitamin D_3_ increases the expression of TRPV6, calbindin-D_9k_, NCX1, and claudins 2/12; consequently, transcellular and paracellular transport of Ca is increased. Effects of 1,25-dihydroxyvitamin D_3_ on pre-duodenal absorption of Ca remain to be elucidated. (**F**), increased concentration of H^+^ stimulates prostaglandin (PG) E synthesis by osteoblast, which stimulates receptor activator of nuclear factor κβ ligand (RANKL) synthesis in adjacent osteoblasts. Additionally, PTH and 1,25-dihydroxyvitamin D_3_ stimulate RANKL expression and suppress the expression of osteoprotegerin (OPG). Reduced OPG allows RANKL to bind to RANK on osteoclast precursors and promote the maturation of those cells. Mature osteoclasts are stimulated by 1,25-dihydroxyvitamin D_3_ promoting the expression of carbonic anhydrase II (CAII), which produces bicarbonate and H^+^ from water and carbon dioxide. The increased H^+^ is secreted into the bone lacunae which facilitates collagen degradation by cathepsin K releasing Ca^2+^ from bone. Within each panel, positive and negative symbols indicate downstream stimulation and repression, respectively.

**Figure 3 animals-14-01232-f003:**
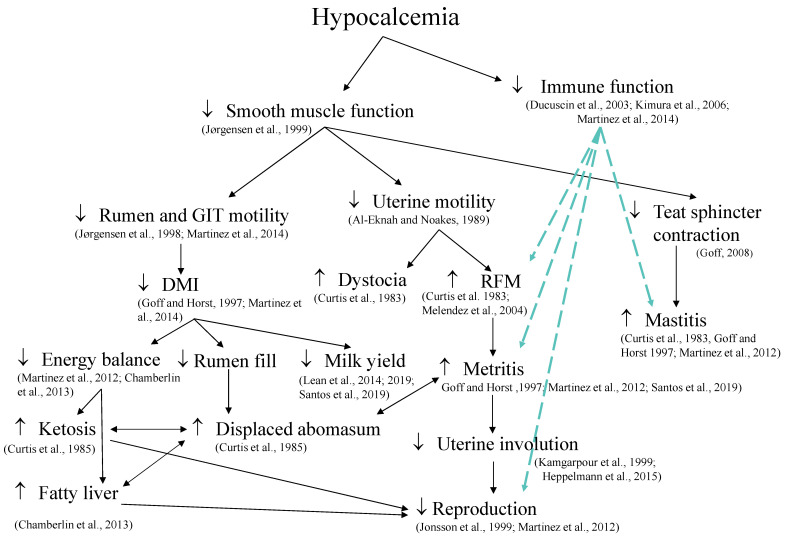
Diagram depicting the associations among diseases in which hypocalcemia has been shown to play a central role either through epidemiological studies or mechanistic experiments. References are included [13,14,16,31,44,147,148,151,152,153,154,155,156,157,158,159,160,161,162]. Continuous lines depict links among diseases. Dashed blue lines depict proposed mechanisms linking immune function and some diseases. Arrows before each disease indicate if hypocalcemia is associated with an increase (arrow up) or decrease/delay (arrow down) in the particular response. DMI = dry matter intake; GIT = gastrointestinal tract; RFM = retained fetal membranes.

## Data Availability

Not applicable.

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
