# Peer review of "Periparturient Mineral Metabolism: Implications to Health and Productivity"

_animals, 2024, doi:10.3390/ani14081232_

Round 1
Reviewer 1 Report
Comments and Suggestions for Authors
General Comments
A nice review article on minerals in transition cows.
Since you mention other preventative measures – should calcium boluses/drenches/ gels be included?
For mechanistic purposes- should the work of Hernandez lab on serotonin/ PTHrP axis be included? Maybe not so critical for day 1 of lactation but maybe has a role after day 4??
What do you think of partial milking- once a day for first 2-3 days? to reduce calcium load?? Some mention of massive dosing of vitamin D or for that matter 25-OHD as preventatives of milk fever. How did they work.
Suggestions
Line 28 …diets marginal to deficient in…
87. Possibly because phosphate absorption is dependent more on rumination (salivary recycling of P) and consuming the diet (even mastectomized cows do not eat while in labor)
104 …at the apical membrane of rumen epithelial syncitia (Remember the rumen epithelium is stratified squamous so it acts as a syncitium of cells with gap junctions)
109 , TRPV6 is primarily, but not exclusively, responsible for…
110 I don’t think diffused is quite right. I believe the evidence suggests the carrier protein calbindin is pulled from one side of cell to the other via cytoskeleton pathways. Check me on that.
111. Instead of carrier protein stay with calbindin all the way .
132 new paragraph after kidney [22].
138 I am afraid I missed your point here. Re-write or eliminate
185/186 …and. In mouse models, increased blood P…
199 “decreased” instead of smaller
207 “Appropriate” instead of adequate
233 Are you sure it is the bound fraction of 25-OHD that goes thru glomerulus- I thought it was the unbound
236 ……. Synthesis or is further hydroxylated ata the 24 and 26 carbons to produce metabolites that are presumably on the catabolic pathway.
241. When you look at the Schroder and Breves work – much of it in Ussing chambers- be cognizant of the types of calcium used to raise Ca concentrations in “rumen lumen fluid” . If transport is dependent on ionized calcium then use of calcium chloride etc is going to make it seem that we reach concentrations that allow paracellular transport in rumen. But what happens in dairy rations where the bulk of the calcium may be coming from limestone???
249 suggest you stick with pre-duodenal instead of pre-intestinal
251 this is likely because there was little 1,25 (OH)2D being produced under conditions that allowed significant rumen Ca absorption.???
254 … between “adjacent” epithelial …..
257 …and these are formed
257 remove IN and replace with Within tight junctions, claudin…
268 cell adhesion protein remove the
278 ..rumen epithelia in Ussing chambers showed…
297 ….100 nM, and the electrical potential (inside cell negative), Ca transport…
301 Suggest adding something like Ca moving from the cytosol to intercellular space is against both the concentration and electrical gradient and therefore requires an input of energy in the form of ATP.
304. Is the Na/Ca exchanger truly more important than the PMCa1 ? Or is it a matter of being constitutively expressed at some level. I know PMCa is upregulated by 1,25D- but is the Na/Ca exchanger??? Am I right in thinking the Na/Ca exchanger is used mostly when the animal is in positive Ca balance- in part to keep cytosolic calcium levels from getting too high- and PMCa used more when animal is in negative Ca balance??? Please inform us readers as to what the literature suggests.
317 This is very apparent in monogastrics such as man and the horse- but is this as true in ruminants, who basically evolved in lower phosphorus environments???
332 /333 Suggest Panel A, Cations, such as Ca++ and H+ can become ionically bound to negatively charged proteins such as albumin. During….
337 You state during metabolic acidosis the parathyroid increases secretion of PTH. Is this referenced?? If ionized Ca is higher – as you state above- then what is the stimulus for higher PTH secretion - thinking of the Ca sensing receptor that is thought to regulate PTH secretion. I suppose the loss of Ca in urine could be a factor if diet Ca was marginal- but how often does that happen??
354 Are you sure 1,25-D suppresses OPG?
357 Can bone release Ca in response to physical change in H+ as a part of body’s buffer system. This can be independent of 1,25D influence Think of Strong ion difference theory and how a change in 0.1 mM Ca in blood might buffer a similar amount of H+. And H+ is normally kept at 0.0001mM!!!
Ie. Bone is a heck of a buffer!!!
367. remove but differently from vitamin D3.
371 how much of this is because the 25-OHD vitamin D2 metabolite does not bind to vitamin D binding protein of blood as well and may not have the same half life?? In chickens for instance they can absorb vitamin D2 just fine – but since it does not bind to their vitamin D binding protein, it is rapidly excreted (or placed into egg yolk- the trick behind high vitamin D eggs by the way).
386 mention that 3 mg Vitamin D is 120,000 IU so people who are not familiar with the vitamin D conversions don’t have to look it up
389. Remember that vitamin D is stored in liver (and adipose – though this may not be as useful), and 25-OHD is not stored to any great extent. Evolutionarily, most ruminants would have a summer to develop good stores of vitamin D, but then in winter with little sun UVb, they would be reliant on vitamin D stores, not 25-OHD stores.
423 Schroder and Breves also published that the rumen DID increase Ca absorption in response to 1,25D – right??? But did not use the same mechanisms.
440 /441 But bone Ca resorption can also occur due to PTH alone (without vit D). As in vitamin D deficient animals???
453 . 1,25 D will generally increase urine Ca when administered by itself – as it raises blood Ca levels. I think if you look carefully , in vivo studies, 1,25-D may be synergistic with PTH to decrease urine Ca. But PTH alone is the primary determinant of Ca reabsorption in distal tubules???
475/490 Is it truly hypocalcemia if ionized Ca is in normal range? Please take a look at this paper by Hisaeda et al. J. Vet. Med. Sci. 82(4): 457–462, 2020. It appears the drop in total calcium measured in most studies is due to a concurrent drop in albumin concentration- part of an acute phase response. This drops total Ca but not ionized. Hence when I as a young vet gave cows with toxic mastitis a bottle of calcium IV to combat low total Ca concentration – I should not have been too surpised when I killed them. The human literature has also observed that sepsis patients have low blood toal calcium – and further, they observe higher mortality when given IV Calcium???
513 acute phase protein response more important than leaking??? I suspect there are papers that would allow you to calculate how much albumin leaked into the milk of a cow with mastitis? Volume X concentration in milk. Compare it to total albumin in blood???
520/524 How many of these studies actually measured both ionized and total Ca
By the way – this review could do us all a favor if it looked at HOW people are measuring ionized Ca??? So many of the handheld cartridge type rely on a drop of whole blood placed on cartridge- how does exposure to atmospheric air change ionized Ca? How do they correct for pH – (and do they correct for pH) compared to the old more reliable blood gas analyzer type machines that gave ionized Ca values????? My gut feeling is that a good deal of what is published is not reliable due to the ionized calcium meters being used – they are not all accurate.
545 – is this reference 133 not 131. Actually – as I continue looking at the papers cited I think you got off by 1 or 2 in your numbering system from this point on!!
573 In vitro nay not be whole story - Better check out the human literature suggesting in vivo that giving calcium to raise blood calcium is detrimental!!
593/594 I think literature would suggest the major replenishment of Ca to endoplasmic reticulum is from cytosolic pols – re-uptake of the Ca released by endoplasmic reticulum in first place - NOT form extracellular calcium, which does happen but is a minor pathway.
639 …addition of inorganic acids to preserve the forages. In the Scandinavian countries the sugar content did not allow for good fermentation and it rains so much dry hay was impractical. In the 1930’s Finnish scientist Artur Virtanen used HCl and H2SO4 to decrease pH of ensiled forages (grasses). This advanced agriculture so much he received the Nobel prize in chemistry in 1945 for this work. So there is hope for some young agricultural scientist to get the Nobel!!!
681 - keep in perspective a one pH unit change in blood pH would be a dead cow. So in vivo this effect is much smaller. Life is permissible between 7.2 and maybe 7.525??
683 If high Ca is fed does metabolic acidosis cause pre-partum PTH levels to rise – or 1,25-OH2D either? Some studies suggest this but most do not.
685 that paper suggests improved PTH responsiveness NOT increased PTH receptor expression. I cannot think of any paper that has looked at PTH receptor content of cell membranes during acidosis -especially in cows??
689 isn’t this purely physicochemical – ie. I did not think PGE was necessary to this action??
696 – how much of this response was due to the fact that Cacl2 was used to acidify the cow and the non-acidified cows were fed calcium carbonate – with lower bioavailability? The increased Ca absorption was likely due to increased 1,25D expression following PTH secretion??!!
720 does this number change if you feed a lot of calcium carbonate (alkalinizing??)
725 .. and improved response …
740 suggest …. lead to a very small reduction in blood calcium concentration. This small change in blood Ca from the cow’s Ca-sensing receptor “set-point” causes secretion of large amounts of PTH, which in turn activates …..
CHECK REFERENCE NUMBERS!!!!
758/765 Some would say the Ca bound to silicate is released upon reaching the abomasum – and then perhaps recombines in small intestine where pH is higher? Or is this more of a problem for the Mg ions or trace minerals bound this way.
772 - And how important is dry matter intake pre-partum. Discuss. Take a look at Goldhawk paper and Huzzey papers that suggest more prevalence of ketosis ???
777- maybe need to reference NASEM 2021 to be more up to date?//
778 – 0.5 mg / day = 20,000 IU – use term nutritionists are more familiar with.
783- how much variation in year is due to sun exposure in dairy – do we know? Has 25-OHD changing from time of Horst to time of Nelson an indication of overfeeding of vitamins???
787/790 Read Jones G Am J Clin Nutr. 2008 Aug;88(2):582S-586S. doi: 10.1093/ajcn/88.2.582S. PMID: 18689406. At near toxic levels 25-OHD will also act as a ligand for VDR. These higher doses of 25-OHD do bring blood levels to levels similar to those found in the old papers of cows with vitamin D toxicity. The difference is likely that these are safer due to shorter half life of straight 25-OHD vs Vitamin D plus the induced 25-OHD
788- That study is mostly reporting an effect of the anionic diets. A number of papers dating back to the 1970’s have tried to utilize 25-OHD as a means of preventing milk fever and hypocalcemia and none showed an effect by itself. I would be careful suggesting this is a valuable tool for improving periparturient Ca status.
794- how repeatable was this increased milk production??? I see this number used by some sales people – is follow up coming??? I do think Nelson makes some good points about this compound as an immune modulator, which I do think you should describe more thoroughly in this section.
Reference list is all off by 1-2 or more??
815 - But did it improve Ca on day 1 after calving when milk fever is most likely? See Hove reference – actually giving Ca alone on day 1 made things worse than if no 1,25-D was given. The paper of Vieria-Neto (wrong ref. number) worked well on day 2 ,3 etc but the anion diet fed was the secret that kept cows from developing milk fever on day 1??? This is where you need to discuss how long it would take for a shot of 1,25- (OH)2 D to activate gut Ca absorption or bone Ca release. Could it be made faster?
821 suggest … period of transient…..
Comments on the Quality of English LanguageNone besides those mentioned above
Author Response
Responses in the attached file labeled 1. ANIMALS-2902403 Responses to Reviewer 1

Reviewer 2 Report
Comments and Suggestions for Authors
This manuscript reviews very nicely the available literature about hypocalcemia, Ca homeostasis, and strategies to prevent hypocalcemia in transition dairy cows. It provides interesting insights and a very thorough analysis of the topic. I only have some minor comments for the authors to consider.
L9. Please remove the running head
L68 What is considered as low blood Ca?
L74. It appears that transient hypocalcemia can be associated with greater milk yield. Consider the McArt and Neves reference.
Figure 1 and 2. I don’t know if it was a conversion issue, but the resolution of the figures is not so high. Please try to improve it.
L128. Please replace ) with ]
L128-132. Are there some cow data related to this?
L136-146. This paragraph is not so easy to read and redundant. Consider rephrasing.
L207. Could the authors provide suggested dietary contents?
L224-226. Excessive BCS increases also inflammation. It could be related to chapter 4.
L464-471. Not only diseases induce inflammation, but also inflammation can cause diseases. Please consider it.
L516. Alkalosis has not been introduced before. It is not immediately clear the connection with albumin.
L746-749. This is a repetition of L54. It can be summarized and just the daily need reported.
L773. Low P diets are mentioned but not discussed. They are relevant to the topic and it might be worth discussing about them.
L777. Consider using NASEM requirements?
L784. Could the authors report the exact quantity?
L800. What does “predisposed to hypocalcemia” mean?
L838. Low Ca and P diets should be mentioned in the conclusions.
Author Response
2. ANIMALS-2902403 Responses to Reviewer 2
